# BOUNDED ATTACKS AND ROBUSTNESS IN IMAGE TRANSFORM DOMAINS

## ABSTRACT

Classical image transformation such as the discrete cosine transform (DCT) and the discrete wavelet transforms (DWTs) provide semantically meaningful representations of images. In this paper we propose a general method for adversarial attacks in such transform domains that, in contrast to prior work, obey the $L^\infty$ constraint in the pixel domain. The key idea is to replace the standard attack based on projections with the barrier method. Experiments with DCT and DWTs produce adversarial examples that are significantly more similar to the original than with prior attacks. Further, through adversarial training we show that robustness against our attacks transfers to robustness against a broad class of common image perturbations.

## 1 INTRODUCTION

Adversarial attacks Biggio et al. (2013); Szegedy et al. (2014); Papernot et al. (2016a) have raised concerns about the safety and robustness of deploying neural networks in critical decision-making processes. Given a neural network that makes accurate predictions on clean data, these attacks modify inputs in a way indiscernible to humans to produce erroneous predictions.

Adversarial attacks can be broadly grouped into black-box and white-box Papernot et al. (2016a); Tramèr et al. (2018). White-box attacks have full access to the neural network architecture, its weights, the training data and the learning algorithm Goodfellow et al. (2015); Kurakin et al. (2017); Papernot et al. (2016a); Madry et al. (2018); Croce & Hein (2020). Black-box attacks are only allowed to perform queries on the target network and observe the input-output relationship Narodytska & Kasiviswanathan (2017); Brendel et al. (2017); Su et al. (2019); Andriushchenko et al. (2020). Many approaches have been proposed to detect adversarial examples Xu et al. (2018); Ma et al. (2018); Feinman et al. (2017); Metzen et al. (2017) and defend against them Gu & Rigazio (2014); Papernot et al. (2016b); Liao et al. (2018); Xie et al. (2019); Zhou et al. (2021). However, most of these defenses can again be broken by suitable adaptive attacks Tramèr et al. (2020); Carlini & Wagner (2017).

Adversarial training Kurakin et al. (2017); Madry et al. (2018), a seminal approach that augments the training data with adversarial examples, reveals to be effective in training empirically Zhang et al. (2019) and provably Salman et al. (2019) robust neural networks. Another approach proposed by Balunovic & Vechev (2019) combines adversarial training with provable defenses to boost the certified robustness. Further, the robustness of trained neural networks can be verified formally through abstract interpretations and relaxations Singh et al. (2019); Xu et al. (2020); Bunel et al. (2020); Müller et al. (2022).

Typically, the distance between a clean and a perturbed input is measured by an $L^p$ norm [1]. In particular, $L^0$(a pseudonorm) and $L^\infty$ have been argued to be necessary adversarial robustness metrics for images Kotyan & Vargas (2022) since they are easily interpretable: number of modified pixels and pixel-wise threshold, respectively. Further, Hendrycks & Dietterich (2019) noticed an interesting interaction between the $L^\infty$ adversarial robustness and common image corruptions such as motion blur, shot noise, and frost. An additional argument for $L^2$ and $L^\infty$ perturbations are the closed formulas for projections needed in common attacks like the Projected Gradient Descent (PGD) Shafahi

---

[1]All norms in this paper are vector norms, i.e., an $H \times W$ RGB image is considered as vector in $\mathbb{R}^n$ $n = 3HW$, not as a matrix.

et al. (2019); Wong et al. (2020); Madry et al. (2018). A different set of techniques aims to perturb in semantically more meaningful ways, e.g., by inserting a carefully chosen patch into the image Thys et al. (2019); Zolfi et al. (2021); Eykholt et al. (2018). The high level idea of bringing image processing knowledge to the problem also motivates our contribution explained next.

**Motivation and Contributions**   Images are not random grids of pixels but can be approximately modeled as first-order Gauss-Markov random fields, which enables JPEG compression. Concretely, when decomposed into frequencies by discrete cosine transforms (DCT) Rao & Yip (2001) at the heart of JPEG or the hierarchical discrete wavelet transforms (DWT) Daubechies (1992) for JPEG 2000, most of the norm concentrates around the low frequencies, which is a key characteristic of images. Prior work have used some of these transforms as a defense to attenuate the additive perturbation noise injected by adversarial attacks Das et al. (2017); Guo et al. (2018), or as a form of data augmentation Duan et al. (2021); Hossain et al. (2019). Furthermore, perturbations in the transformed domain were used to defend against pixel attacks Bafna et al. (2018) or to carry attacks in the transformed domains Duan et al. (2021); Hossain et al. (2019); Deng & Karam (2020); Shi et al. (2021a); Luo et al. (2022). However, these attacks did not bound the effect of the change in the pixel domain.

The goal of our work is to provide adversarial attacks in transform domains, which thus can exploit their expressiveness, while at the same time obeying the common $L^\infty$ bounds in the pixel domain. Doing so makes the amount of change interpretable, enables comparison to prior attacks, and leverages the interaction with various common image corruptions Hendrycks & Dietterich (2019). The challenge is in the high-dimensional geometry, which makes it difficult to derive the projections needed in PGD-based attacks and thus a different approach is needed. Specifically, we contribute:

- A novel white-box attack based on the *barrier method* from nonlinear programming that does not require any closed-form projections and can be instantiated for a large class of transforms. Our focus is on DCT and DWTs.

- An evaluation of our attacks against prior work on ImageNet. In particular, given the same $L^\infty$ bound, we show that our attacks consistently yield adversarial examples with significantly higher similarity to the original, as verified by the Learned Perceptual Image Patch Similarity (LPIPS) metric. As a baseline we also include a hand-crafted PGD-based attack for DCTs to illustrate the challenges in obtaining projections that obey $L^\infty$ bounds.

- Adversarial training using the adversarial examples produced by our attack run on ImageNet. We show that the obtained networks provide better robustness against common image corruptions on CIFAR-10 spanning different categories including noise, blur, weather and digital corruptions.

## 2   THE ATTACK FORMULATION

Many image transforms have been invented to provide expressive representations of images Rao & Yip (2001). Widely used examples include the DCT and DWTs at the heart of the JPEG and JPEG 2000 compression standards Wallace (1992); Adams (2001). Both are linear and invertible and decompose an image into a notion of frequencies, in which high frequencies capture details that often can be removed with little visual impact. An example is shown in Fig. 1. We aim to leverage such expressive transform representations for adversarial attacks while, in contrast to prior work, obeying the widely used $L^\infty$ box defined in the pixel space. Our approach is applicable to a large number of transforms; thus, we first present it in a general way before instantiating it to DCT and DWTs.

Let $\boldsymbol{x}^0 \in [0,1]^n$ be a clean image correctly classified as $c$ by a classification model $f$ (pixel color channel values are assumed normalized to $[0,1]$). $l$ is a loss function, for example a cross-entropy. Let $\phi$ be an invertible and differentiable image transform that maps the original image from the pixel space to a domain with a desirable expressiveness. In this $\phi$-domain, we seek to find a perturbed version $\boldsymbol{y}'$ of $\boldsymbol{y}^0 = \phi(\boldsymbol{x}^0)$ such that $\boldsymbol{x}' = \phi^{-1}(\boldsymbol{y}')$ gets misclassified and $\left\| \boldsymbol{x}' - \boldsymbol{x}^0 \right\|_\infty \le \epsilon$ in the pixel domain. $\boldsymbol{y}'$, and thus $\boldsymbol{x}'$, can be computed by solving the following constrained optimization problem:

$$\min_{\boldsymbol{y}} -l(\phi^{-1}(\boldsymbol{y}), c) \text{ subject to } \left\| \phi^{-1}(\boldsymbol{y}) - \boldsymbol{x}^0 \right\|_\infty \le \epsilon. \tag{1}$$

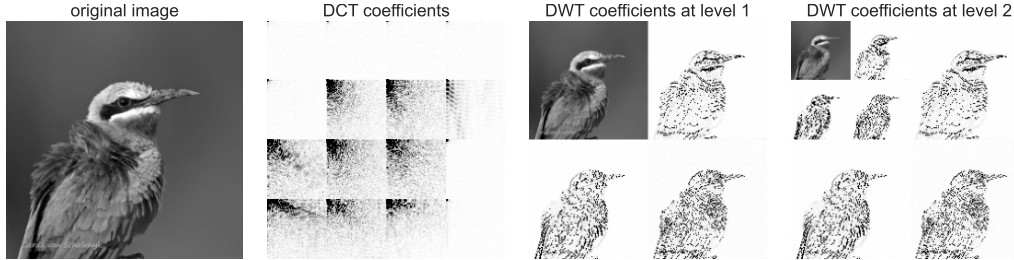

Figure 1: Example of a block DCT and multi-level DWT decompositions (in our work we use up to five levels) of an ImageNet sample in grayscale. In the transform domains, magnitude is encoded by darkness, with white being zero. The DCT concentrates the values in each block in the top left, the DWT recursively decomposes into a downscaled version of the image plus horizontal and vertical details at different levels.

In principle, this problem can be resolved by a projected gradient descent (PGD) scheme, analogous to Madry et al. (2018). This amounts to iterating over two phases:

$$z^{t+1} = y^t + \eta \operatorname{sign}(\nabla_y l(\phi^{-1}(y^t), c)), \tag{2}$$

$$y^{t+1} = P^*(z^{t+1}, \epsilon). \tag{3}$$

Equation 2 minimizes the inverse loss function. It requires computing the gradient of the loss with respect to the input $\nabla_y l(\phi^{-1}(y^t), c)$, taking its sign, and moving in the opposite direction with a step size $\eta$. Equation 3 projects on the $L^\infty$-box defined in the pixel space but operates in the $\phi$-domain. The derivation of such a $P^*$ is challenging since $\phi$ is not $L^\infty$-preserving: $\left\| x - x^0 \right\|_\infty \neq \left\| \phi(x) - \phi(x^0) \right\|_\infty$. Given $\phi$ it may be possible to hand-craft a suitable $P^*$. We do this later in Section 4 for the DCT to show the issues involved. However, our main contribution is an approach different from PGD that eliminates the need for such projections and is applicable to a large set of transforms with minor modifications.

## 3 THE BARRIER METHOD

In this section, we propose a method entirely different from the PGD approach from Madry et al. (2018) to solve problem 1. It is based on the so-called barrier method Nocedal & Wright (2006) from nonlinear programming. In the context of adversarial attacks, the barrier method was used before by Finlay et al. (2019) to enforce a decision boundary constraint, which is fundamentally different from the box constraint in the transform domain that we are targeting.

At the heart of barrier methods is the *barrier function* that incorporates the inequality constraints in conjunction with the objective function in a way that minimizing yields a solution to the original problem. Formally, problem 1 is solved as

$$\min_y -l(\phi^{-1}(y), c) - \mu \log(\epsilon - \left\| \phi^{-1}(y) - x^0 \right\|_\infty). \tag{4}$$

The formulation depends on the free parameter $\mu$ which controls the balance between the loss term $-l(.,.)$ and the logarithmic term that embeds the box constraint. The log term can be seen as a smooth approximation of the indicator function that is $= 1$ if $\left\| \phi^{-1}(y) - x^0 \right\|_\infty \leq \epsilon$ is fulfilled and $= 0$ otherwise. It serves as a penalty since it grows very large near the boundary of the box.

A straightforward approach computes the gradient that is zero for all dimensions expect the one dimension that yields the maximum absolute value. For instance, if $x^t = (-2.5, 1.5)$ with $L^\infty(x^t) = 2.5$ the gradient will be $\nabla_x L^\infty(x^t) = (-1, 0)$. This causes slow convergence since only one dimension is updated per iteration. Fig. 2a shows an illustration of the barrier function in two dimensions. The log term $-\mu \log(\epsilon - \left\| \phi^{-1}(y) - x^0 \right\|_\infty)$ smoothly enforces the constraint $\max_i |\phi^{-1}(y)_i - x^0_i| \leq \epsilon$ that is also satisfied when $|\phi^{-1}(y)_i - x^0_i| \leq \epsilon$ for all $i$. Using the latter condition with the log barrier method in each dimension we obtain the formulation that we use

$$\min_y -l(\phi^{-1}(y), c) - \mu \sum_{i=0}^{n-1} \log(\epsilon - |\phi^{-1}(y)_i - x^0_i|). \tag{5}$$

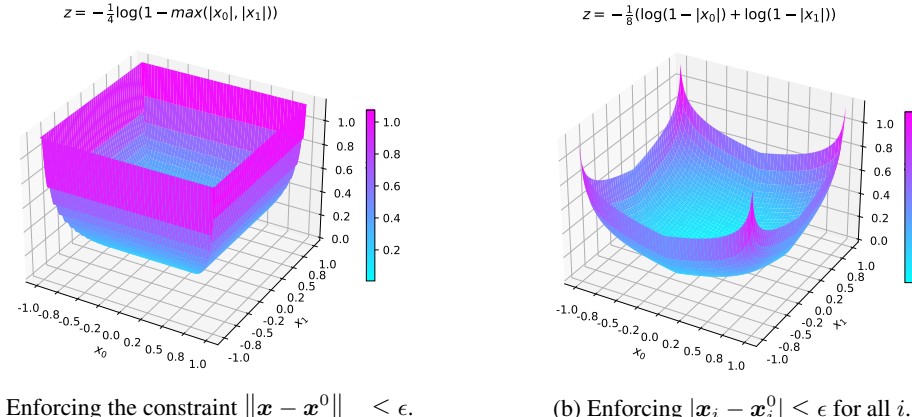

(a) Enforcing the constraint $\left\| \boldsymbol{x} - \boldsymbol{x}^0 \right\|_\infty \leq \epsilon$.

(b) Enforcing $|\boldsymbol{x}_i - \boldsymbol{x}_i^0| \leq \epsilon$ for all $i$.

Figure 2: A two dimensional illustration of the continuous functions enforcing the $L^\infty$ box constraint for $\epsilon = 1$, $\boldsymbol{x}^0 = (0, 0)$, $l = 0$, and $\phi$ is the identity function.

A two dimensional illustration of this barrier function is shown in Fig. 2b.

After this continuous relaxation of the discrete condition, our new formulation does not need further projections and can be solved by a generic standard gradient descent-based optimizer. Just like the original PGD attack, there is no formal guarantee of convergence for an arbitrary classifier $f$. In practice, we obtained the best results using the difference of logits ratio loss function as loss $l$ proposed by Croce & Hein (2020) while being optimized by Adam Kingma & Ba (2015).

In this paper, we instantiate the barrier method for $L^\infty$-preserving attacks in the domains of the DCT and several DWT families, applied up to five levels. For DWTs, we leverage in addition their semantics by preserving the down-scaled version of the original image in the top-left corner (see Fig. 1) in our attacks.

## 4 PROJECTION APPROXIMATION FOR THE DCT

We briefly sketch the challenges in deriving a suitable projection operator $P^*$ in equation 3 to enable a PGD-based attack in a transform domain, by doing so for the DCT. The general problem is that commonly used transforms are not $L^\infty$-preserving and thus $P^*$ has to be derived specifically from the transform definition using high-dimensional geometry, which may even become infeasible.

Let $\mathbb{H}$ be the $L^\infty$ box of radius $\epsilon$ defined in the pixel space around a given clean image $\boldsymbol{x}^0$: $\mathbb{H} = B_\infty(\boldsymbol{x}^0, \epsilon)$. Applying DCT preserves the volume (due to orthogonality) but rotates $\mathbb{H}$. As a result, the maximal $L^\infty$ norm of $\mathrm{dct}(\mathbb{H})$ is substantially larger than $\epsilon$. Thus when projecting $\boldsymbol{z}^{t+1}$ in equation 2, we have to ensure that again that the inverse $\mathrm{idct}(P^*(\boldsymbol{z}^{t+1}, \epsilon)) \in \mathbb{H}$. To this end, we define $P_*$ as a standard $L^\infty$ projection (clipping) but the box we are projecting onto must have an adequate width. This radius is given by the following lemma. The proof is technical and based on the definition of the DCT as a linear transform whose matrix consists of cosine values. Details about the definition and the proof are in the supplements B and C.

**Lemma 1.** *Let* $\kappa = (1 + 2\sqrt{2} \cdot \psi + 2 \cdot \psi^2)/N$ *with* $\psi = -1/2 + \sin(N + \frac{1}{2})\frac{\pi}{2N}/2\sin(\frac{\pi}{4N})$. *Then*

$$\mathrm{idct}(B_\infty(\boldsymbol{y}^0, \epsilon/\kappa)) \subseteq B_\infty(\boldsymbol{x}^0, \epsilon),$$

*and there is no smaller* $\kappa$ *with this property.*

The situation is sketched in Fig. 3 in two dimensions $N = 2$ (instead of 64 as for the JPEG standard). On the left we have $\boldsymbol{y}^t$ and $\boldsymbol{x}^t$ in DCT- and pixel-space, respectively. A step of size $\eta$ in direction of the gradient with maximal loss is determined to obtain $\boldsymbol{z}^{t+1}$. On the right $\boldsymbol{z}^{t+1}$ is projected onto $\mathrm{idct}(B_\infty(\boldsymbol{y}^0, \epsilon/\kappa))$ to obtain $\boldsymbol{y}^{t+1}$ and ensure that $\boldsymbol{x}^{t+1}$ is in $\mathbb{H}$, which is in general in the interior, in contrast to the original PGD attack.

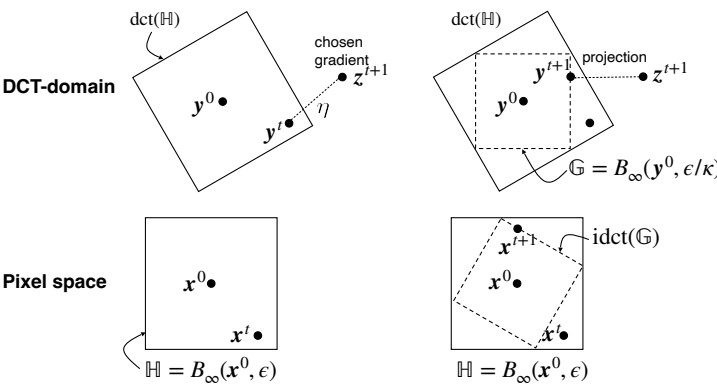

Figure 3: One iteration of PGD in DCT space. The figure serves as intuition for the effect of rotations on the $L^\infty$-norm by considering only 2 (instead of $8 \cdot 8 = 64$) dimensions.

## 5 EXPERIMENTAL EVALUATION

In this section, we first evaluate our attacks in the DCT and DWT transform domains and compare to prior attacks. Then we investigate their use for adversarial training and, in particular, the robustness of the obtained networks against common image corruptions. All of our code and scripts to reproduce the experiments will be made available as open source.

We evaluate our attacks in Section 5.2 with images from ImageNet using a trained vision transformer architecture Dosovitskiy et al. (2020). In Section 5.2, for adversarial training and evaluation of robustness towards common image corruptions, we used a smaller ConvNet model trained on CIFAR-10. Its architecture consists of 16 convolutional layers, 5 max pooling layers, and 3 fully connected layers. Convolutions are followed by batch normalization and ReLU activation functions. The fully connected layers are preceded by dropout layers. This architecture is similar to that used by Zhu et al. (2021).

### 5.1 COMPARISON OF THE DIFFERENT ATTACKS

We compare our attacks against four baselines: standard PGD, the two variants of the automatic projected gradient descent attack (APGD) Croce & Hein (2020) and the square attack Andriushchenko et al. (2020). All attacks have the same budget of $k = 100$ iterations. The step size of the standard PGD attack is set to $\eta = \epsilon \times \frac{5}{2k}$. The other attacks we invoke as implemented in the *AutoAttack* suit Croce & Hein (2020). We discuss the instantiation and the parameterization of our attacks in more detail next.

**Instantiation of our attacks**  Our barrier attack from Section 3 is instantiated for the DCT in a way compatible with JPEG compression pipeline, and called `dct_barrier`. This means, first we perform a color conversion from RGB to YCrCb[2] before the DCT is applied block-wise on $8 \times 8$ blocks.

Similarly, `dwt_barrier_l` is the instantiation of our attack compatible with JPEG-2000, where we consider two types of wavelets: Cohen-Daubechies-Feauveau (CDF) 9/7 and LeGall 5/3, applied with $l$ levels. Unless noted otherwise, the default is CDF 9/7 with one level. The parameter $\mu$ in equation 5 is initially set to $\mu = 10/n$ ($n$ is the dimension of the image) and is doubled to further strengthen the log term if the constraint is violated or having an underflow problem. We minimize the barrier functions with the Adam optimizer Kingma & Ba (2015) with an initial step size rate of 0.01 and the default beta parameters of $(0.9, 0.999)$.

---

[2]The conversion from RGB to YCrCbA is a pixel-wise affine transformation. Further details can be found in the supplement A.

| | | Our attacks | | | | | Baseline attacks | | |
|---|---|---|---|---|---|---|---|---|---|
| | dct_pgd | dct_barrier | dwt_barrier_1 | dwt_barrier_3 | dwt_barrier_5 | pgd | apgd-ce | apgd-dlr | square |
| $\epsilon = 0.02$ | | | | | | | | | |
| $L^\infty$ | 0.02 | 0.0126 | 0.0113 | 0.013 | 0.0129 | 0.02 | 0.02 | 0.02 | 0.018 |
| $L^2$ | 1.5702 | 0.1842 | 0.107 | 0.1917 | 0.1986 | 4.3551 | 5.795 | 5.513 | 6.9304 |
| similarity distance | 0.0013 | 0.0001 | 0.0 | 0.0002 | 0.0002 | 0.0265 | 0.0575 | 0.0568 | 0.0403 |
| success rate (%) | 63.74 | 26.97 | 7.39 | 28.77 | 30.17 | 99.9 | 100.0 | 100.0 | 90.21 |
| $\epsilon = 0.03$ | | | | | | | | | |
| $L^\infty$ | 0.03 | 0.024 | 0.0211 | 0.0237 | 0.024 | 0.03 | 0.0251 | 0.0251 | 0.0249 |
| $L^2$ | 2.1926 | 0.611 | 0.3083 | 0.5689 | 0.5942 | 5.4517 | 6.89 | 6.8642 | 9.5579 |
| similarity distance | 0.003 | 0.0012 | 0.0004 | 0.0012 | 0.0012 | 0.0446 | 0.094 | 0.1014 | 0.0771 |
| success rate (%) | 80.1 | 75.12 | 39.8 | 69.15 | 77.11 | 100.0 | 100.0 | 100.0 | 99.5 |
| $\epsilon = 0.1$ | | | | | | | | | |
| $L^\infty$ | 0.0978 | 0.0817 | 0.0838 | 0.0811 | 0.0807 | 0.1 | 0.1 | 0.1 | 0.1 |
| $L^2$ | 5.2214 | 3.1043 | 2.4656 | 2.8278 | 2.8493 | 11.9267 | 24.8942 | 27.2576 | 37.5832 |
| similarity distance | 0.0157 | 0.0208 | 0.0147 | 0.0174 | 0.0178 | 0.166 | 0.4398 | 0.5122 | 0.3527 |
| success rate (%) | 99.0 | 99.4 | 93.61 | 99.8 | 99.3 | 100.0 | 100.0 | 100.0 | 100.0 |
| $\epsilon = 0.2$ | | | | | | | | | |
| $L^\infty$ | 0.166 | 0.1567 | 0.1639 | 0.1533 | 0.1531 | 0.2 | 0.1672 | 0.1672 | 0.1672 |
| $L^2$ | 7.6775 | 6.2671 | 5.3749 | 5.3251 | 5.3982 | 20.0369 | 40.1668 | 45.4996 | 60.5106 |
| similarity distance | 0.0369 | 0.0739 | 0.057 | 0.0558 | 0.0582 | 0.3327 | 0.6578 | 0.7498 | 0.5161 |
| success rate (%) | 100.0 | 100.0 | 100.0 | 100.0 | 100.0 | 100.0 | 100.0 | 100.0 | 100.0 |
| $\epsilon = 0.3$ | | | | | | | | | |
| $L^\infty$ | 0.2057 | 0.2286 | 0.2387 | 0.2244 | 0.2237 | 0.3 | 0.2522 | 0.2522 | 0.2522 |
| $L^2$ | 9.8083 | 9.1894 | 7.9604 | 7.9484 | 7.8762 | 28.3351 | 59.038 | 66.9566 | 88.0086 |
| similarity distance | 0.0628 | 0.1352 | 0.1072 | 0.1104 | 0.1101 | 0.4717 | 0.8383 | 0.92 | 0.6805 |
| success rate (%) | 100.0 | 100.0 | 99.5 | 100.0 | 100.0 | 100.0 | 100.0 | 100.0 | 100.0 |

Table 1: Evaluation of our four proposed attacks alongside with five baseline attacks for multiple $L^\infty$ box radii $\epsilon$. We show the average $L^\infty$ and $L^2$ norm of the retrieved adversarial examples, the success rate of the attacks and the similarity distance that these adversarial images have with respect to the clean images using the LPIPS metric (lower values mean better similarity). The success rate is over 1000 test images randomly sampled from ImageNet, and the distance metrics are means.

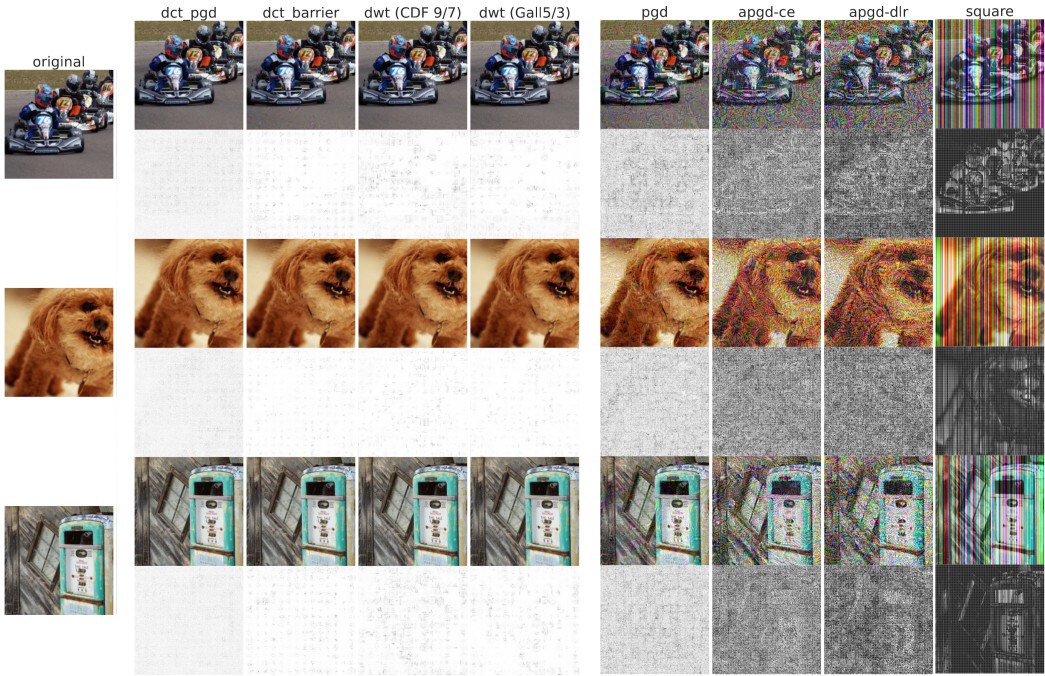

Figure 4: An illustration of different adversarial examples found by our proposed attacks compared to the baseline all run with $L^\infty$ box radius value of $\epsilon = 0.3$. Each row of adversarial images is followed by a row of heatmaps representing the pixelwise difference w.r.t the corresponding clean image in the first column. In these, white $= 0$ and black $= \epsilon$.

Our PGD-based attack presented from Section 4 is called dct_pgd. The standard PGD attack uses as step size parameter $\eta$ the size $\epsilon$ of the box multiplied by $\frac{5}{2k}$. Taking into account our shrinkage factor of $\kappa$ induced by the DCT (Lemma 1) and a second scaling factor $\rho = 2.772$, caused by the color conversion from RGB to YCbCr, we choose $\eta = 5\epsilon/(2\kappa\rho k)$.

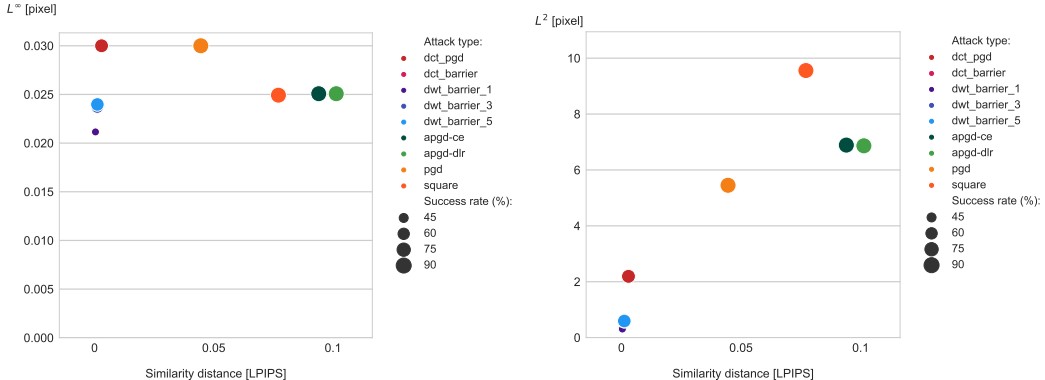

Figure 5: Visualization of a part of the results presented in Table 1. The two plots correspond to a targeted $L^\infty$ box of radius $\epsilon = 0.03$. The success rate is encoded with the marker size.

**Results**   We ran all the mentioned attacks on a sample of 1000 correctly classified test images and show the mean of the results in Table 1. For three randomly selected images, we provide a visualization of the adversarial examples and their difference to the original in Fig. 4. In addition, we plot a subset of this tables's entries that correspond to $\epsilon = 0.03$ in Fig. 5 to show the interplay of $L^\infty$-bound, LPIPS similarity distance, and attack success rate.

From Fig. 4, we notice visually that the adversarial examples found are significantly less modified compared to those reported by the pixel-based attacks, even when targeting relatively large $L^\infty$ box radii, while still using almost the full slack provided by the $\epsilon$ bound. Table 1 confirms this higher visual similarity by a consistently higher similarity in the LPIPS metric across all experiments. The trade-off for this higher similarity is in a lower success rate for very small $\epsilon$, whereas the prior benchmarks almost always succeed. For example, for the DCT, run for $\epsilon = 0.03$, it drops to about 75%. For the DWT with one level it is then only 39%, which is likely due to us freezing the scaled-down version of the original image in the top left corner (see Fig. 1), which effectively freezes one quarter of all DWT values. Indeed for decompositions with more levels the success increases significantly to 69% (`dwt_barrier_3`) and 77% (`dwt_barrier_5`) while still maintaining substantially small similarities distance in contrast to the attacks operating in the pixel domain. For the even smaller $\epsilon = 0.02$, the trend to lower success rates continues. Thus, for very small $\epsilon$ our DCT and DWT attacks are not suited.

We remark that the adversarial examples retrieved by the `dct_pgd` have, on average, an $L^\infty$ norm smaller than the radius of the targeted box. In other words, these attacks, in most cases, do not necessarily reach the boundary of the box, unlike `pgd` which always ends up at this extreme. This phenomenon is most likely due to the rotation caused by DCT which leads to the elimination of a part of the search space that we conjecture contains more noisy images on which the `pgd` attack lands.

**Analysis of the barrier-based attack against PGD**   As we have seen above, the barrier-based attacks usually do not reach the boundary of the $L^\infty$-box. To analyze this behaviour, we also implemented a barrier-based attack in the pixel domain, called `barrier`. We performed an experiment tracking the evolution of the $L^\infty$ norm, and also the $L^2$ norm, of the explored images throughout the iterations of the `barrier` and the `pgd` attacks targeting an $L^\infty$ box of radius 0.2. The results are shown in Fig. 6. The `pgd` attack is likely to reach the $l_\infty$ box boundary in the first few iterations due to the frequent projections on the boundary. The barrier method, in contrast reaches the boundary only slowly due to the logarithmic term and usually not entirely. This term encourages the attack to explore the vicinity of the original image before moving away towards the boundary. The comparison of the $L^2$ norms show that the overall noise introduced by the barrier method is smaller.

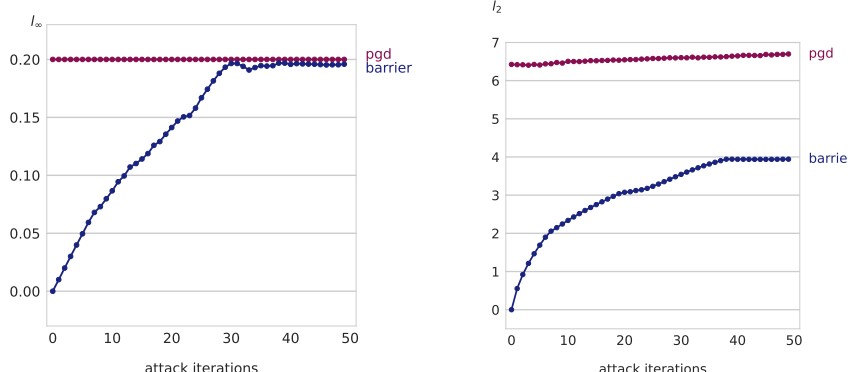

Figure 6: Comparison between our `barrier` attack and the standard `pgd` by tracking the evolution of the norm value through the iterations.

## 5.2 Adversarial training

The high similarity of the adversarial examples produced by our attacks motivates their use as proxies to achieve robustness against yet another class of image perturbations that also preserves visual similarity: common image corruptions. To this end, we use our attacks for adversarial training (AT), a technique in which the neural network is trained on adversarial examples aiming to increase robustness against this adversary. We train for robustness against two adversaries: the standard PGD and our DCT-based attack. We choose to focus on the `dct_pgd` variant given its identical computational cost to the standard PGD. Both attacks are granted the same number of iterations $k = 30$ with step sizes identical to those discussed in Section 5.1. For each adversary, we perform the adversarial training targeting three $L^\infty$ box radii. This means that in total six ConvNets (of the same architecture outlined earlier) are shown in Table 2.

**Adversarial robustness** First, we report the natural accuracy of these networks. Then, we evaluate their robustness against the corresponding adaptive attack that is used for training and their robustness against the other attack. The results in Table 2 show that the neural networks trained with PGD in the RGB space (standard AT) have significantly lower natural accuracy, in contrast to the networks trained with our attack in the DCT space (AT with DCT). For large values of $\epsilon$, the accuracy of standard adversarial training remains low around 10 %, which is indistinguishable to random predictions given that the dataset contains 10 classes. In contrast, our adversarial training largely preserves the natural accuracy. We notice that the robustness against the DCT attack does not imply robustness against the PGD attack.

**Adversarial robustness transferability towards common image corruptions** We consider the CIFAR-10-C dataset Hendrycks & Dietterich (2019), a benchmark constructed by applying common image corruptions to the CIFAR-10 test set. These corruptions are only used for evaluation and not to augment the data during training. There is a near perfect correlation between the adversarial accuracy in the DCT space and the accuracy associated with these corruptions. In other words, evaluating a neural network using our attack offers a good approximation of the accuracy against a variety of common image corruptions. More importantly, the results shown in Table 2 show that the adversarial robustness against our proposed DCT-based attacks transfers. Neural networks trained using our DCT-based attacks are found to always be more robust to common image corruptions across all categories noise, blur, weather and digital. This means the expressiveness of the DCT representation is well-suited to increase robustness against these corruptions.

## 6 Related work

We provide more details on prior work on robustness related to discrete image transforms. Most aimed at defending against pixel-based adversarial attacks or improving the generalization of neural networks towards common image corruptions. The work of Dziugaite et al. (2016); Das et al.

| | Natural | Adversarial | | Noise | | | Blur | | | | Weather | | | | Digital | | | |
|---|---|---|---|---|---|---|---|---|---|---|---|---|---|---|---|---|---|---|
| | | RGB | DCT | Gauss | Shot | Impulse | Defocus | Gauss | Motion | Zoom | Snow | Fost | Fog | Bright | Contrast | Elastic | Pixel | JPEG |
| $\epsilon = 0.03$ | | | | | | | | | | | | | | | | | | |
| Standard AT | 63 | 39 | 60 | 61 | 62 | 59 | 60 | 58 | 57 | 58 | 60 | 55 | 43 | 62 | 33 | 59 | 61 | 62 |
| AT with DCT (ours) | 85 | 5 | 80 | 77 | 79 | 73 | 77 | 73 | 70 | 74 | 78 | 78 | 70 | 83 | 55 | 76 | 81 | 83 |
| $\epsilon = 0.1$ | | | | | | | | | | | | | | | | | | |
| Standard AT | 26 | 20 | 24 | 25 | 25 | 24 | 26 | 26 | 26 | 26 | 24 | 20 | 23 | 23 | 22 | 26 | 26 | 26 |
| AT with DCT (ours) | 80 | 0 | 70 | 79 | 79 | 76 | 74 | 72 | 68 | 72 | 74 | 71 | 59 | 76 | 47 | 72 | 78 | 79 |
| $\epsilon = 0.2$ | | | | | | | | | | | | | | | | | | |
| Standard AT | 10 | 10 | 10 | 10 | 10 | 10 | 10 | 10 | 10 | 10 | 10 | 10 | 10 | 10 | 10 | 10 | 10 | 10 |
| AT with DCT (ours) | 76 | 0 | 62 | 76 | 76 | 75 | 71 | 69 | 66 | 68 | 70 | 65 | 51 | 72 | 41 | 68 | 74 | 75 |

Table 2: A summary of the accuracies of different neural networks trained with either the standard PGD (standard AT) or our DCT attack (AT with DCT) for multiple $\epsilon$ box radii.

(2017); Guo et al. (2018) aims to filter out noise from the adversarial examples by adjusting various quality factor values during JPEG compression/decompression, which amounts to reducing the magnitude of the DCT coefficients. Closely related, Bafna et al. (2018) sought $L^0$ robustness through projecting the largest DCT coefficients. These defenses has been shown to be breakable through adaptive attacks Shin & Song (2017); Tramèr et al. (2020), specifically, by approximating the non-differentiable rounding operator of the JPEG compression and running a gradient-based attack. Other fast and iterative rounding schemes have been proposed in Shi et al. (2021b). Yin et al. (2019); Guo et al. (2019) considers $L^2$ perturbations that preserve norms due to orthogonality of the used transforms, discrete Fourier transform (DFT) and DCT respectively.

The work in Duan et al. (2021) generates adversarial examples by removing information in the DCT domain. The $L^\infty$ box used is on the JPEG quantization matrix instead of the input image. Since the DCT coefficients of the clean image are element-wise divided by this matrix before rounding, larger box radii allow their technique to eliminate more frequencies from the image. In the same direction, Hossain et al. (2019) preceded the neural network by a DCT based layer that randomly crops some DCT coefficients during training. This can be interpreted as an extension of the dropout technique aiming at its regularization effects. Yahya et al. (2020) propose a gradient-free method that obtains adversarial examples by mixing the frequencies of a clean image with the frequencies of another auxiliary image that they call watermark. In addition to FFT and DCT, they make use two wavelets: Haar and Daubechies 3. Sharma et al. (2019) applies masks to selectively perturb low and high frequencies. Much like Deng & Karam (2020); Shi et al. (2021a), all these works do not provide any guarantee on the bounds in the pixel space, which is the primary contribution in our work.

Luo et al. (2022) explicitly uses a similarity distance in the optimization problem formulation in the pursuit of semantically similar adversarial examples. In our work, we only enforce an $L^\infty$ box and still achieve a high semantic similarity that is not explicitly used in the optimization. Finlay et al. (2019) invokes the barrier function on the loss while minimizing the norm in a manner similar to Carlini & Wagner (2017) without accounting for the box.

Finally, the work in Gueguen et al. (2018); dos Santos & Almeida (2021) proposed neural network architectures that operate directly on the JPEG format to avoid decompression before classification. Our work could be used to study the robustness of such architectures.

## 7 CONCLUSION

We proposed the first method to adversarially attack image classification networks in transform domains while observing an $L^\infty$-bound in the pixel domain. Our approach is based on the barrier method, which eliminates the need for projections and can thus be easily instantiated for a large class of transforms, linear and beyond. The results with DCT and DWTs show the relevance of our attacks: we obtain adversarial examples that are more similar to the original than prior work. Further, when used for training, the obtained robustness transfers to common image perturbations, which demonstrates the expressiveness of the considered transforms. We see our contribution as a step towards leveraging decades of research on image representations for better robustness.

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

## A  COLOR CONVERSION

YCbCr is a color system where each pixel is represented by three values: the brightness Y and a pair encoding the of color spectrum CbCr. Let $c$ the RGB triplet of a pixel in the position $(i, j)$. Then $d$, the YCbCr values at this same pixel, are given by $d = A \cdot c + b$, where

$$A = \begin{pmatrix} 0.299 & 0.587 & 0.114 \\ -0.1687 & -0.3312 & 0.5 \\ 0.5 & -0.4186 & -0.0813 \end{pmatrix} \text{ and } b = \begin{pmatrix} 0 \\ 128/255 \\ 128/255 \end{pmatrix}.$$

## B  THE DEFINITION OF DCT

Let $X$ be a 2D block of an image of size $N \times N$. The DCT transform of $X$ is a matrix $Y$ of size $N \times N$ where the coefficient at the position $(p, q)$ is given by:

$$Y_{pq} = \frac{\alpha_p \alpha_q}{N} \sum_{i=0}^{N-1} \sum_{j=0}^{N-1} X_{ij} \cos \frac{(2i+1)p\pi}{2N} \cos \frac{(2j+1)q\pi}{2N}, \quad \alpha_0 = 1 \text{ and } \alpha_r = \sqrt{2}, \ 0 < r < N. \tag{6}$$

## C  PROOF OF LEMMA 1

*Proof.* Let $x$ and $x'$ be two pixel images in some color system. $x$ and $x'$ are subdivided into blocks $X$ and $X'$ of size $N \times N$. $Y$ and $Y'$ are the DCT block coefficients of $X$ and $X'$, respectively. $Y$ and $Y'$ are blocks of the DCT transforms $y = \text{dct}(x)$ and $y' = \text{dct}(x')$. It is clear that if $y' \in B_\infty(y, \epsilon')$ then $Y' \in B_\infty(Y, \epsilon')$. Now, let find the smallest $\epsilon'' > 0$ such as $X' \in B_\infty(X, \epsilon'')$.

$$\begin{aligned}
|X_{ij} - X'_{ij}| &= \frac{1}{N} \left| \sum_{p=0}^{N-1} \sum_{q=0}^{N-1} \alpha_p \alpha_q (Y_{pq} - Y'_{pq}) \cos \frac{(2i+1)p\pi}{2N} \cos \frac{(2j+1)q\pi}{2N} \right| \\
&\leq \frac{1}{N} \sum_{p=0}^{N-1} \sum_{q=0}^{N-1} \alpha_p \alpha_q |Y_{pq} - Y'_{pq}| \left| \cos \frac{(2i+1)p\pi}{2N} \cos \frac{(2j+1)q\pi}{2N} \right| \\
&\leq \frac{1}{N} \sum_{p=0}^{N-1} \sum_{q=0}^{N-1} \alpha_p \alpha_q \epsilon' \left| \cos \frac{p\pi}{2N} \cos \frac{q\pi}{2N} \right| \\
&\leq \frac{\epsilon'}{N} \sum_{p=0}^{N-1} \sum_{q=0}^{N-1} \alpha_p \alpha_q \cos \frac{p\pi}{2N} \cos \frac{q\pi}{2N} \\
&\leq \frac{\epsilon'}{N} \left( \alpha_0^2 + \sum_{q=1}^{N-1} \alpha_0 \alpha_q \cos \frac{q\pi}{2N} + \sum_{p=1}^{N-1} \alpha_0 \alpha_p \cos \frac{p\pi}{2N} + \sum_{p=1}^{N-1} \sum_{q=1}^{N-1} \alpha_p \alpha_q \cos \frac{p\pi}{2N} \cos \frac{q\pi}{2N} \right) \\
&\leq \frac{\epsilon'}{N} \left( 1 + 2\sqrt{2} \sum_{p=1}^{N-1} \cos \frac{p\pi}{2N} + 2 \left( \sum_{p=1}^{N-1} \cos \frac{p\pi}{2N} \right)^2 \right) = \frac{\epsilon'}{N} \left( 1 + 2\sqrt{2}\psi + 2\psi^2 \right) = \epsilon' \kappa
\end{aligned}$$

Now, let's simplify $\psi = \sum_{p=1}^{N-1} \cos \frac{p\pi}{2N}$.

$$\sum_{p=0}^{N} e^{i\frac{p\pi}{2N}} = \frac{e^{i\frac{(N+1)\pi}{2N}} - 1}{e^{i\frac{\pi}{2N}} - 1}$$

$$= e^{i\frac{\pi}{4}} \frac{e^{i\frac{(N+1)\pi}{4N}} - e^{-i\frac{(N+1)\pi}{4N}}}{e^{i\frac{\pi}{4N}} - e^{-i\frac{\pi}{4N}}}$$

$$= \left(\cos\frac{\pi}{4} + i\sin\frac{\pi}{4}\right) \frac{2i\sin\frac{(N+1)\pi}{4N}}{2i\sin\frac{\pi}{4N}}$$

$$= \cos\frac{\pi}{4} \times \frac{\sin\frac{(N+1)\pi}{4N}}{\sin\frac{\pi}{4N}} + i\sin\frac{\pi}{4} \times \frac{\sin\frac{(N+1)\pi}{4N}}{\sin\frac{\pi}{4N}}$$

$$= \frac{2\cos\frac{\pi}{4}\sin\frac{(N+1)\pi}{4N}}{2\sin\frac{\pi}{4N}} + i\sin\frac{\pi}{4} \times \frac{\sin\frac{(N+1)\pi}{4N}}{\sin\frac{\pi}{4N}}$$

$$= \frac{\sin(N+\frac{1}{2})\frac{\pi}{2N} + \sin\frac{\pi}{4N}}{2\sin\frac{\pi}{4N}} + i\sin\frac{\pi}{4} \times \frac{\sin\frac{(N+1)\pi}{4N}}{\sin\frac{\pi}{4N}}$$

$$= \frac{1}{2} + \frac{\sin(N+\frac{1}{2})\frac{\pi}{2N}}{2\sin\frac{\pi}{4N}} + i\sin\frac{\pi}{4} \times \frac{\sin\frac{(N+1)\pi}{4N}}{\sin\frac{\pi}{4N}}$$

, whence

$$\psi = \sum_{p=1}^{N-1} \cos\frac{p\pi}{2N} = Real\left(\sum_{p=0}^{N} e^{i\frac{p\pi}{2N}}\right) - 1$$

$$= -\frac{1}{2} + \frac{\sin(N+\frac{1}{2})\frac{\pi}{2N}}{2\sin\frac{\pi}{4N}}.$$

So we have found $\epsilon'' = \epsilon'\kappa$. To show that this the smallest $\epsilon''$ (and thus smallest $\kappa$) with this property, it suffices to set $\mathbf{Y}'$ at a corner of the hypercube, for instance $\mathbf{Y}'_{pq} = \mathbf{Y}_{pq} - \epsilon'$ for all $p$ and $q$. Then we follow this same derivation that gives equalities instead of inequalities.

Finally, since $\mathrm{idct}(B_\infty(\mathbf{Y}, \epsilon')) \subseteq B_\infty(\mathbf{X}, \epsilon'\kappa)$ holds for all the corresponding blocks $\mathbf{Y}$ and $\mathbf{X}$, this property also holds for the totality of the image $\boldsymbol{x}$ and its transforms $\boldsymbol{y}$: $\mathrm{idct}(B_\infty(\boldsymbol{y}, \epsilon')) \subseteq B_\infty(\boldsymbol{x}, \epsilon'\kappa)$. $\qquad\square$

