# OpenReview forum: "Bounded Attacks and Robustness in Image Transform Domains"
_ICLR.cc/2023/Conference — Submitted to ICLR 2023_

### Official Review · Reviewer_LwcG · 2022-10-23

**Confidence:** 4
**Correctness:** 3
**Technical Novelty And Significance:** 2
**Empirical Novelty And Significance:** Not applicable
**Recommendation:** 5

**Clarity, Quality, Novelty And Reproducibility:**

The paper is mostly well written other than some hiccups ( a couple of examples given in the previous section). The authors also show there novel contribution when adopting the barrier method in the given scenario. In the main paper, the authors do not provide enough detail to completely reproduce the experimental results however. My main question is with respect to contribution, what would be the specific case where dct_barrier and dwt_barrier would be used?

**Strength And Weaknesses:**

Strengths
1. The authors propose the barrier method which allows image perturbations to take place in the transform space while still being in the L-inf bounding box. This approach allows the authors to generate such perturbations for both DCT and DWTs.
2. Experimental results show that even though the success rate of the proposed method is lower than regular attacks (in particular for smaller epsilon values), the generated images are more visually similar to the original unperturbed images.
3. The authors also show that generate perturbed images can be useful for adversarial training as well.


Weaknesses
1. The generated adversarial images are very similar to the original ones. Furthermore, we see that actual L-inf norms are much smaller than the epsilon boundary. Given this is the case, my question is why do we bother with L-inf bounding box for the barrier methods? Shouldn't dct_pgd be enough when we only care about similarity? Given that there are multiple works that show that L-p bounds are not representative of similarity, I am unsure of the purpose of using it in the barrier method.
2. As mentioned in the last point, there are multiple works that show L-p norms are not good in terms of visual similarity. The authors might consider citing such works. Furthermore, as the goal is similarity matching the authors might want to consider works that try to keep the visual similarity in mind when comparing.
3. Table 1 caption reads: "Evaluation of our four proposed attacks alongside with five baseline attacks for...". But in the table, dct_pgd is listed under "Our attacks". This should be part of the baselines if I am not mistaken? Also there is no mention of dct_pgd before this table.
4. Table 2 compares dct_barrier only against standard pgd. This is a weak comparison in particular given the absence of dct_pgd as a baseline here.

**Summary Of The Paper:**

This paper tackles the problem of generating adversarial images in the image transformation domain. In particular the authors are interested in the transformations used in JPEG compressions: discrete cosine transform (DCT) and discrete wavelet transforms (DWTs). While doing so, they still maintain the usual condition used in adversarial image generation which is to bound the perturbation in an L-inf domain. This is not straight forward in the transform domain as the norms do not translate directly. Thus the authors develop a barrier method to mitigate the issue. The authors then run experiments on the imagenet dataset to show the effectiveness of the proposed approach. The results show that even though the success rate of the barrier method can be lower than the other baselines in particular for lower epsilon bounds, the generate adversarial images are more similar to the original image.

**Summary Of The Review:**

The authors provide a novel way to generate adversarial images in the transform domain while still being inside L-inf bounding boxes. The results show that the generated adversarial images are more similar to the original images when compared to attacks on the pixel space. However, my main concern about this paper is the use case. If similarity is something that the authors care the most about then the bounding box should also be on the inverse of this similarity score. This takes away hugely from the contribution of this work in my opinion.

---

### Official Review · Reviewer_GeiG · 2022-10-24

**Confidence:** 3
**Correctness:** 3
**Technical Novelty And Significance:** 2
**Empirical Novelty And Significance:** 2
**Recommendation:** 3

**Clarity, Quality, Novelty And Reproducibility:**

The paper is mostly clear, and the authors say they will make their code available. The proposed novel aspect of this paper is the retention of the constraint in the original input space when solving for the perturbation in the transformation space.

**Strength And Weaknesses:**

Strengths
- The authors show that their proposed attacks result in perturbed images that are closer to the original image while still resulting in high attack success rate for large values of epsilon (radius of the l-infinity norm constraint).
- The paper is fairly straightforward to read, and includes helpful visualizations.
- The proposed method is potentially generalizable to other types of transformations.

Weaknesses
- The motivation for the paper is not made very clear, as in why this formulation of attack would be preferable to standard PGD. The introduction could be reworked to better incorporate this motivation, as the beginning of the introduction is mostly just a summary of adversarial robustness, much of which could be moved to the related work section.
- Evaluating the different attacks (Table 1) may be more interesting when comparing on an adversarially trained model. The attack success rate for the proposed attacks is also significantly lower than existing attacks for the standard choice of epsilon (0.03).
- The adversarial accuracy seems rather low for standard adversarial training in Table 2? Reporting the average common corruptions accuracy in this table would also be beneficial. The model adversarially trained with the DCT PGD-based method is not robust to standard PGD attacks -- again the motivation for using the proposed approach is not totally clear to me here if still considering the same threat model as standard PGD. The paper also makes the claim that the use of DCT in the proposed attack is what results in improved common corruptions accuracy for the model adversarially trained with DCT, however, it seems likely that the improved common corruptions accuracy is just due to the natural/standard accuracy being much higher for this model as compared to the standard adversarially trained model. To make this claim, it’d be useful to compare to a standard non-adversarially trained model.
- It's unclear to me how generalizable this is to other types of transformations given the projection approximation required.

Additional comments/suggestions:
- The readability may be improved if sections 2 and 4 are rearranged to be sequential.
- Table 1 is a little small.
- Make use of parenthetical citations (\citep) when appropriate for better readability.
- The use of the variable y as the transformed input throughout (i.e. in Equation 1) is somewhat confusing due to the usual choice of y to represent the ground truth label.

**Summary Of The Paper:**

This paper introduces an l-infinity norm bounded adversarial attack that operates in some transformation space rather than input space. In particular, the authors use the discrete cosine transform (DCT) or the discrete wavelet transform (DWT) to map from the input space to the corresponding domain, solve for the perturbed example in this domain, and map this perturbed example back to the original input space via the inverse transform such that the l-infinity norm constraint is satisfied in the original input space. The authors show that the resulting perturbed image (in the original input space) is closer in distance (according to L2 and LPIPS metrics) to the original image than the perturbed image found using projected gradient descent (PGD) given the same l-infinity norm constraints. The authors provide two different methods for constructing the attack in this manner, one based on PGD and one based on the barrier method. The authors compare their proposed attacks to existing attack methods on a standard trained ImageNet classifier in terms of distance from the original image, and show the results of adversarial training using the DCT PGD-based attack method.

**Summary Of The Review:**

I am leaning towards reject because the motivation for using the proposed attack methods is unclear to me, given that these methods do not seriously compete with standard PGD in terms of attack strength for the typical choice of epsilon. The evaluations of the proposed methods (for both attacking a trained model, and adversarially training a model), could also be more extensive to support the paper's claims, such as the stated improvement on common corruptions.

---

### Official Review · Reviewer_RQTp · 2022-10-25

**Confidence:** 3
**Correctness:** 2
**Technical Novelty And Significance:** 2
**Empirical Novelty And Significance:** 2
**Recommendation:** 3

**Clarity, Quality, Novelty And Reproducibility:**

Limited clarity, quality, novelty, and reproducibility, see [Strength And Weaknesses].

**Details Of Ethics Concerns:**

No ethics concerns.

**Strength And Weaknesses:**

Strength:
- Exploring imperceptible adversarial examples is promising.
- Introducing the barrier method to craft adversarial examples is novel to me.
- Various scenarios are considered in the experimental setting.

Weakness:
- The motivation to introduce an image transformation is unclear.
- The motivation of the method is unclear. It is unclear for me to figure out the connection between Eq. (4) and the barrier function, given Eq. (1). Moreover, it is also unclear why the authors consider the gradient computation after introducing the barrier function. It is unclear to me whether Eq. (5) has the same impact on the effectiveness as a \ell_0 norm.
- The motivation of the experiments is unclear. Specifically, the authors claim that they evaluate the proposed attacks. Built upon it, I cannot figure out which part and which kinds of superiority are supported by the conducted experiments.
- Related works are missing. The authors claim to increase the perceptibility of adversarial examples. A similar work is [1], which designs perceptual adversarial examples. However, this work does not discuss and compare with it.
- The paper is hard to follow. For example, the contribution part is confusing. It is confusing that ‘Our focus is on DCT and DWTs.’ is listed in the contribution. In addition, the authors do not explain the mentioned barrier method before the use in contribution.


Suggestions:
1 I do suggest the author improve the writing.
2 The introduction makes me have to revisit the abstract, as the abstract provides limited information about the work, and the relation to the introduction is weak.

[1] Perceptual adversarial robustness: Defense against unseen threat models.


**Summary Of The Paper:**

This work aims to propose a novel adversarial attack method. The authors propose introducing a barrier method to generate adversarial examples. Then, the authors conduct some experiments.

**Summary Of The Review:**

The paper's motivation, method, and experiments are unclear.

---

### Official Review · Reviewer_2yuU · 2022-10-28

**Confidence:** 3
**Clarity, Quality, Novelty And Reproducibility:** The paper is written clearly. My main…
**Correctness:** 3
**Technical Novelty And Significance:** 2
**Empirical Novelty And Significance:** 1
**Recommendation:** 3

**Strength And Weaknesses:**

Strengths:
The paper is easy to follow and the proposed method is explained clearly.

Weaknesses:
My main concerns are with the experiments. Details listed below.

1. From the experiments, it is unclear what advantage the proposed methods have over regular PGD. The authors claim that their method produces images of better quality and do not violate the l_infty contraint. But you can do this easily with PGD by clipping to satisfy some l_infty constraint. Table 1 shows that the proposed method has a weaker success rate compared to PGD, but has a better visual quality. However, the proper way to compare would be to plot the curves of "visual-distortion" (measured by PSNR, LPIPS, or L_infty) vs "attack success rate" for both methods. Currently we are just looking at individual points on the two curves and have no way of distinguishing which method is better.

2. Why DCT? The paper claims DCT is for JPEG compatibility, but I do not see any evaluations where the attacked images are post-processed through JPEG or JPEG 2000.

**Summary Of The Paper:**

This paper proposes a method for white-box adversarial attacks which is bounded in l_infinity constraint. The method uses a barrier formulation together with DCT formulation.

**Summary Of The Review:**

Due to concerns over the experiments, I recommend rejection.

---

### Decision · Program_Chairs · 2023-01-20

**Decision:**

Reject

**Justification For Why Not Higher Score:**

Reviewers are uniformly unclear about the motivation for the work and whether this actually an advance. My read of the paper confirms these concerns.

**Justification For Why Not Lower Score:**

N/A

**Metareview: Summary, Strengths And Weaknesses:**

The paper introduces an L-infinity norm bounded adversarial attack that operates in DCT/DWT space rather than in pixel space, and uses the inverse DCT/DWT transform to obtain the corresponding adversarial image. It considers two instantiations of this attack: one based on PGD and one based on a barrier method.

The motivation for using the proposed attack rather than “vanilla” PGD is unclear. The paper claims the proposed attack produces fewer visible perturbations than PGD (as shown in Figure 4) but the success rate of the attack at low epsilon values is also substantially worse than that of PGD (per Table 1) so the comparison in Figure 4 doesn’t seem like an apples-to-apples comparison. As a result, it remains unclear if the attack presented in the paper actually is a substantial advance over prior art.